# Evolution of the Road Network Topology of Central European Housing Estates

Pál Hegyi , Attila Borsos * and Csaba Koren

Department of Transport Infrastructure and Water Resources Engineering, University of Győr, Egyetem tér 1, 9026 Győr, Hungary; hegyip@sze.hu (P.H.); koren@sze.hu (C.K.)
* Correspondence: borsosa@sze.hu

**Abstract:** The analysis of road network topology has attracted the attention of researchers in the past few decades. In this study, the road topology of housing estates in a few selected Central European countries (Hungary, Austria, Czech Republic, and Slovakia) was analysed. This research was carried out in three steps: (1) the road network topology of different decades from the 1950s to the 1980s was described, (2) the ratio of intersections and dead-ends was investigated, and (3) the connectivity indices were analysed and compared. The research was carried out using ESRI ArcGIS software. The results show that the design of road networks built in different countries is similar in the housing estates studied. When analysing the road networks over time, significant differences could be found in the case of Hungary for housing estates built after the 1960s. In general, connectivity has become more important, as it has gradually increased over time.

**Keywords:** housing estate; connectivity; road network topology; Central Europe; intersection type; GIS; transport infrastructure

## 1. Introduction

The investigation of road network topology originates in the US, where districts or entire settlements have developed with a given street pattern. These patterns have evolved naturally over time, but they are to a greater extent are the result of conscious design. In Europe, and in particular in Central Europe such wide-spread areas characterised by a uniform road network pattern have not emerged [1]. Instead, residential districts in smaller sizes having different road patterns were built.

City developments reflect the characteristics of different ages, including how their road networks expanded over time. The morphological characteristics of any district are influenced by block sizes and traversability, land use, and building forms [2], as well as economic and social factors [3].

Typical residential areas in cities are housing estates. Road networks within housing estates show different patterns depending on the era in which they were built and whether they were greenfield or brownfield investments. In the case of the latter, particularly in Central Europe, suburban buildings were replaced with four/ten-storey buildings with modifications to the existing road network. In addition, the motorisation level and design guidelines also had an impact on street network patterns. Among other things, guidelines regulated the number and location of parking places (outdoor or indoor parking facilities). The road network topology and traversability of a district are also influenced by the location of the given district within the city, as well as by its external connections.

In many European countries, the adaptation of the so-called Radburn design (New York, NY, USA) is also visible [4]. The guiding principle of this design approach is to separate motorised and nonmotorised users. To this end, only nonmotorised traffic is allowed inside the block, while motorised traffic operates around the block and can access the buildings through dead-ends.

To describe these processes, we need to quantify various aspects of the road network topology. Road network topology can be described with various indicators [5–7]. These geometric characteristics can be specified using geographic tools and software, for which a nonexhaustive list of which is given below:

- The ratio of road links and nodes [8];
- The ratio of the dead-ends and intersections;
- The degree, which represents the number of links it has to other nodes [9–11];
- The betweenness centrality, which is the number of the shortest paths through a node in the network; this method can be used to study social relationships, public transport, road network topology, and traffic flow possibilities [12–14];
- The closeness centrality, which shows the distance of the investigated point from other points in the network; this is the average length of the shortest paths from the point.

One of the most cited authors in this area of research is Marshall [15], who gave a comprehensive overview of street network topology according to the different eras. He differentiated four basic street network patterns (Figure 1), all of which have their own unique design traits. In reality, these patterns cannot be always purely identified on their own; instead some kind of combination is present. Each type is characterised by its use and design as follows:

- Type A—historic core: typical of the core area of historical cities, it is characterised by an irregular design.
- Type B—gridiron (central, extension, or citywide): typical of planned extensions or new settlements; it is characterised by a regular design.
- Type C—anywhere: including individual villages or suburban extensions, and is perhaps the most general type.
- Type D—peripheral development: common in suburbs in the USA, and often associated with curvilinear street layouts. One of the features is many dead-ends or cul-de-sacs (a circular turnaround at the end with lots radiating outward).

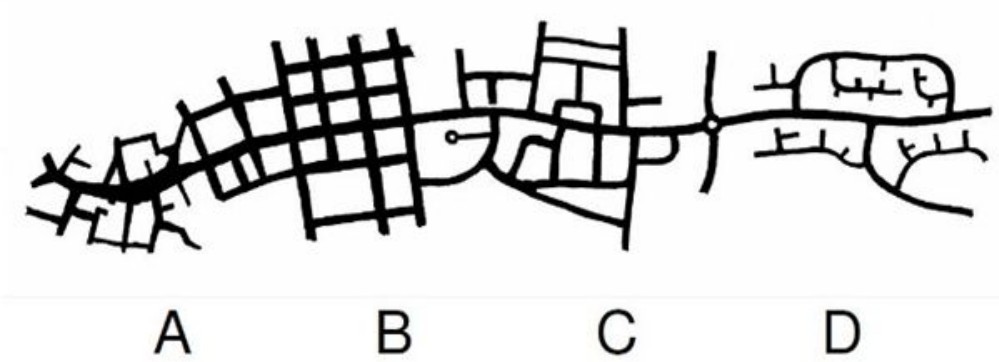

**Figure 1.** Road network patterns according to Marshall [15].

Referring to Figure 1, Han et al. [16] argued that network patterns evolve from grid-like to tree-like and that there are intermediate stages. In their paper, they defined four road network patterns (pure tree-like networks, cul-de-sac networks, T-shaped networks, and grid-shaped networks). Boeing [17] studied 27,000 US urban street networks using a similar classification.

By investigating 168 urban districts in four cities—London, New York, Hong Kong, and Gdansk—Huang et al. [18] shed light on the hidden relationships between street patterns and behaviour, in both the virtual and physical worlds. One of their conclusions was that a compact, well-connected street network promotes urban life and should be encouraged, while cul-de-sac networks and three-way intersections should be discouraged, because they reduce the link-to-node ratio.

Besides theoretical considerations, street network analysis may have practical implications. The authors in Barrington-Leigh and Millard-Ball [19], for example, argued that

street network sprawl leads to excess travel and pollution. The authors in Marshall and Garrick [20] associated street network forms with road safety, whereas Choi and Ewing [21] concluded that the street network do not influence safety, and they instead argued that denser and more connected neighbourhoods have significantly lower congestion levels. The authors in Soltani et al. [22] reasoned that improving the spatial continuity of the local street network can promote active mobility to schools. The authors in Delclòs-Alió et al. [23] hypothesised that car rates and their growth are associated with street networks with low intersection density, but they did not obtain statistically significant results.

In Central Europe, a special type of urban district was created after World War II; in many countries, multistorey housing estates were built to alleviate the housing shortage. This paper analyses the road network structure of these developments. The novelty of the paper lies in the fact that we collected an international sample of housing estates and investigated the development of their road networks over four decades. Such an international analysis of Central European housing estates is not available in the literature.

The methodology is presented in Section 2. Section 3 introduces the study area and the characteristics of the housing estates selected for the study. Section 4 contains results, followed by the discussion and limitations in Section 5. Finally, in Section 6, the main conclusions are summarised.

## 2. Method

Road pattern data come from the OSM (OpenStreetMap) [24] open-source database; data extracts can be downloaded from [25]. The original map is available in the WGS84 geographic coordinate system, which is used in many countries around the world. Map data of each city have been transformed into the local coordinate system used in the given country.

As the OpenStreetMap is maintained by volunteers, there are no clear regulations about how map attributes should be recorded. Therefore, the first step was to clean the original data based on the attributes, where network elements with the attributes 'residential' and 'living-street' were selected. In addition, the selected areas in the urban road networks were also investigated using Google Street View. Network edges which may have different attributes are also part of the investigated network, and were therefore also selected. Turning traffic flows at major intersections may have resulted in additional nodes. Strictly speaking, these are not considered elements of the road network, but as they influence the ratio of the intersection types, they had to be filtered out. The result of this cleaning process is the motorised vehicle road network. Links and nodes of bicycle and pedestrian facilities were not considered. As an example, Figure 2 shows the original database downloaded from OpenStreetMap (left side) and the modified road network (right side) of a housing estate in Győr, Hungary.

Vertices (corner points) and nodes had to be treated separately, because edges in a road network start and end at intersections. Therefore, vertices had to be filtered out of the original database.

In this research, the road network patterns of housing estates were investigated using the two most commonly used indicators [26,27]: the connectivity index and the ratio of dead-ends to intersections. The connectivity index is a measure of the traversability of the road network and can be calculated using the formula below (Equation (1)).

$$connectivity\ index = \frac{road\ edges}{topological\ points}. \tag{1}$$

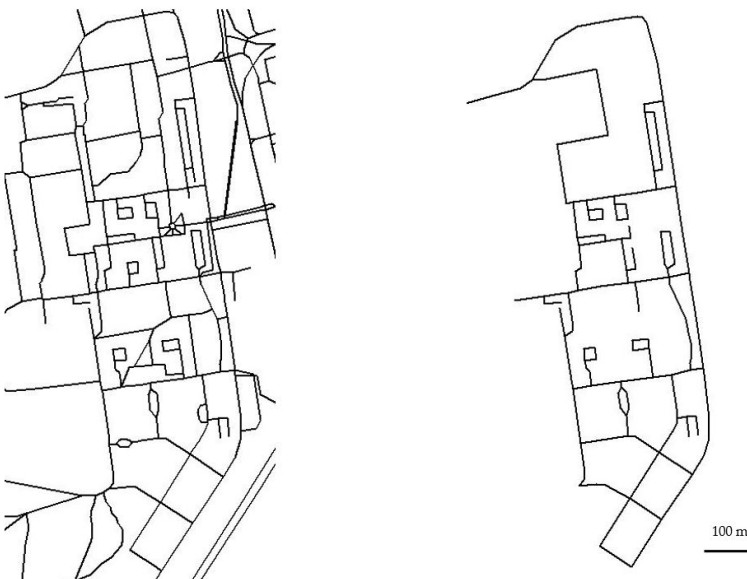

**Figure 2.** Original map of the road network of a district downloaded from OpenStreetMap (**left**) and the modified network (**right**) in the city of Győr.

This index shows the ratio of edges to topological points, where road edges represent the number of links in the network, and topological points represent the number of dead-ends and intersections (or nodes). The definition of dead-ends needs some explanation. Dead-ends can be interpreted on both a macro and a micro level. The two circles in Figure 3 are dead-ends on macro level; however, on micro level, the blue circle is a dead-end, and the red circle is a three-way intersection. In the latter case, such points are identified by the ArcGIS software as nodes with two links instead of three, because there are only two link IDs. Even though these nodes are three-way intersections, they do not contribute to the traversability of the road network and were considered as dead-ends (macro level).

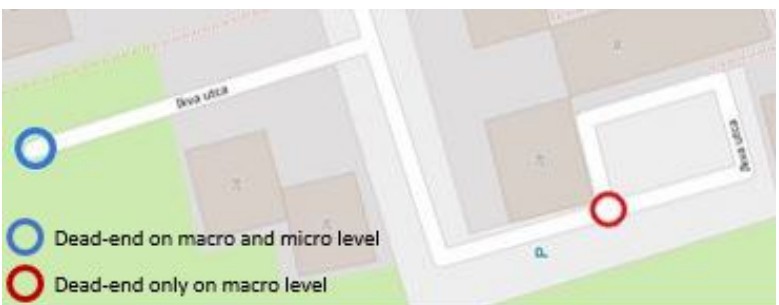

**Figure 3.** Node marked with a red circle is a dead-end on macro level and a three-way intersection on micro level.

As dead-ends were interpreted on macro level, the same approach was used for both major roads and intersections. In the case of roads where the original map showed a double line, only one link was taken into account. Roundabouts were categorised according to the number of legs. Intersections with more than four legs were categorised as four-way intersections; this was a very rare case.

All indicators were calculated on the basis of the internal road network of housing estates. This means that links and nodes on network edges were only considered if they were part of the internal network. This is not the same case as a main road running through the housing estate. The type and year of construction were also taken into account when defining the network boundaries. We analysed the road network using the ESRI ArcGIS software, and we performed the statistical analyses in R [28].

### 3. Study Area

The scope of the current research includes the road network patterns of Central European housing estates built between 1950 and 1990 in Hungarian, Austrian, Czech, and Slovak cities. The populations of these cities ranges from 100,000 to 250,000. A total of 37 housing estates were selected and studied. They are situated in different districts, on the outskirts or in the inner city, and include both greenfield and brownfield developments. Housing estates from the following cities are included in this study: Debrecen, Győr, Kecskemét, Miskolc, Nyíregyháza, Pécs, Szeged, and Székesfehérvár (Hungary); Linz, Salzburg, and Innsbruck (Austria); Plzen, Olomouc, and Liberec (Czech Republic); and Kosice (Slovakia) (Figure 4).

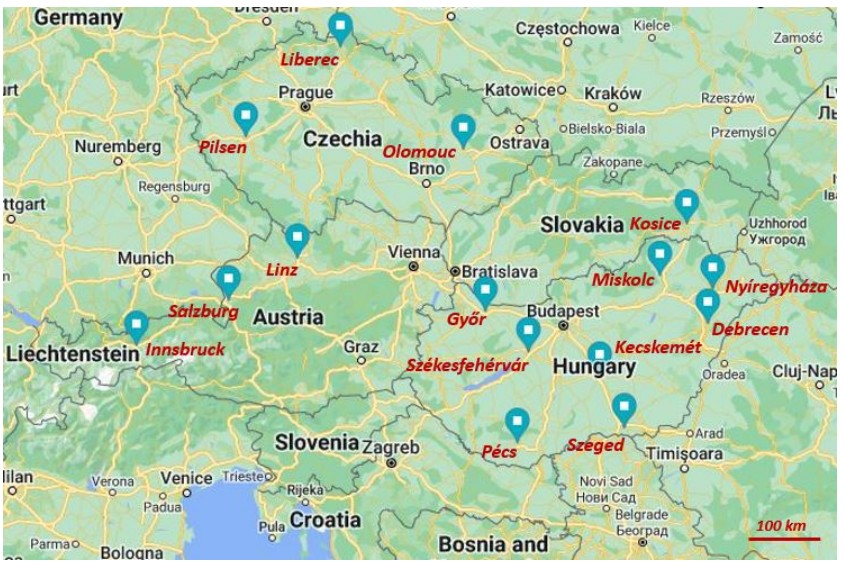

**Figure 4.** Cities from which housing estates were selected for the study.

Tables 1 and 2 show some basic characteristics of Hungarian and nonHungarian housing estates, as well as some derived indicators. Housing estates from Austria, the Czech Republic, and Slovakia were selected from cities to match the population range of the Hungarian cities. In terms of area, housing estates were quite heterogeneous, ranging from 0.1 to 1.9 sqkm. However, the area of most housing estates was between 0.2 and 0.8 sqkm. The reason for the lower limit is that housing estates with a small area and an incomplete road network were not included, as it was not possible to analyse their network topology. At the upper end, housing estates larger than 1.0 sqkm were rarely found, as the amount of internal traffic in these areas usually requires a main road within the district, thus dividing it into two separate parts in terms of this analysis.

Our sample included housing estates from the former "East Bloc" countries, such as Hungary, Slovakia, and the Czech Republic, as well as from Austria, which belonged to the "West". There were undoubtedly differences between these countries in the processes of establishing housing estates; However, these differences are beyond the scope of this research. From this sample, it is remarkable that Hungary has several large housing estates (over 1 sqkm or 10,000 population), whereas the other three countries, with only two exceptions (Innsbruck Reichenau and Kosice KVP), have estates of a smaller size. This latter subset of countries was quite limited in terms of sample size.

The road networks of the selected housing estates were taken from the OSM database, which was based on the actual network data at the time of this research. Based on local experiences, it was assumed that the internal network structure of the housing estates did not change over time. The changes related to the roads in these areas were mostly the expansion of parking capacity and traffic engineering measures, such as signalization on the surrounding arterial roads. These changes did not affect the network structure. However, it must be acknowledged that some minor changes may have occurred.

In relation to the size of the housing estate, the network lengths of the estates studied varied widely. The average length of the road section in between two intersections is more homogeneous between the sites, mostly between 60 and 100 m. The range of the road density between the sites is also not too wide, with most areas having a density of between 10 and 20 km/sqkm.

In terms of land use, Hungarian and Slovak housing estates are quite homogeneous, thereby being predominantly residential. Austrian and Czech estates have mixed land use, with single-family houses, as well as industrial and commercial blocks. These areas were included in the analysis on the basis of the following principles:

- The district boundary should be at least a two-lane main road.
- A lower road category was accepted if the land use pattern on either side was significantly different.
- Natural and artificial boundaries such as rivers, railway lines, etc. were considered as primary boundaries.

**Table 1.** Main characteristics of the Hungarian housing estates studied.

| City | Housing Estate | Area [sqkm] | Population | Network Length [km] | Ratio of Dead-Ends [%] | Ratio of Three-Way Intersections [%] | Ratio of Four-Way Intersections [%] | Average Road Length [m] | Connectivity Index | Road Network Density [km/sqkm] | Intersection Density [Intersection/sqkm] |
|---|---|---|---|---|---|---|---|---|---|---|---|
| Győr | Adyváros | 0.8 | 17,500 | 10.3 | 37.3 | 59.0 | 3.6 | 98 | 1.25 | 12.8 | 65.0 |
| | József Attila | 0.3 | 6560 | 5.3 | 40.0 | 56.9 | 3.1 | 70 | 1.15 | 17.6 | 130.0 |
| | Marcalváros I. | 0.6 | 11,500 | 7.7 | 27.1 | 61.0 | 11.9 | 91 | 1.42 | 12.8 | 71.7 |
| | Marcalváros II. | 0.4 | 6900 | 6.8 | 25.0 | 70.6 | 4.4 | 76 | 1.31 | 17.0 | 127.5 |
| Székesfehérvár | Palotaváros | 0.9 | 16,160 | 17.8 | 25.3 | 69.4 | 5.3 | 78 | 1.34 | 19.7 | 141.1 |
| | Tóváros | 0.2 | 3510 | 4.7 | 38.2 | 55.9 | 5.9 | 59 | 1.18 | 23.5 | 210.0 |
| Pécs | Uránváros | 1.0 | 25,000 | 17.6 | 32.7 | 62.5 | 4.8 | 77 | 1.35 | 17.6 | 113.0 |
| | Kertváros | 1.9 | 35,478 | 32.2 | 22.8 | 68.7 | 8.5 | 88 | 1.41 | 17.0 | 105.3 |
| Miskolc | Avas I. | 0.2 | 5822 | 3.5 | 13.0 | 82.6 | 4.3 | 228 | 1.17 | 17.7 | 100.0 |
| | Avas II. | 0.5 | 12,864 | 10.1 | 39.8 | 47.6 | 12.6 | 97 | 1.12 | 20.2 | 124.0 |
| | Avas III. | 0.5 | 6889 | 10.3 | 35.7 | 51.4 | 12.9 | 103 | 1.44 | 20.7 | 90.0 |
| Debrecen | Dobozi | 0.3 | 5866 | 4.4 | 30.8 | 57.7 | 11.5 | 106 | 1.58 | 14.6 | 60.0 |
| | Libakert (1950s) | 0.2 | 3950 | 3.7 | 21.2 | 75.8 | 3.0 | 76 | 1.45 | 18.3 | 130.0 |
| | Libakert | 0.4 | 4870 | 8.2 | 23.3 | 66.7 | 10.0 | 94 | 1.45 | 20.4 | 115.0 |
| | Tócóskert | 0.8 | 11,118 | 11.6 | 17.4 | 77.9 | 4.7 | 94 | 1.44 | 14.5 | 88.8 |
| | Tócóvölgy | 0.2 | 10,126 | 4.2 | 28.0 | 64.0 | 8.0 | 112 | 1.48 | 20.9 | 90.0 |
| | Újkert | 0.7 | 16,166 | 8.8 | 36.2 | 56.5 | 7.2 | 92 | 1.39 | 12.6 | 62.9 |
| | Vénkert | 0.3 | 8083 | 3.4 | 29.6 | 70.4 | 0.0 | 89 | 1.41 | 11.3 | 63.3 |
| | Wesselényi | 0.1 | 2213 | 2.1 | 14.3 | 71.4 | 14.3 | 93 | 1.57 | 20.6 | 120.0 |
| Nyíregyháza | Örökösföld | 0.5 | 10,420 | 8.3 | 24.3 | 74.3 | 1.4 | 83 | 1.43 | 16.6 | 106.0 |
| Szeged | Odessza | 0.3 | 6200 | 5.4 | 28.8 | 69.2 | 1.9 | 81 | 1.29 | 18.1 | 123.3 |
| | Felsőváros | 1.3 | 15,576 | 23.5 | 28.2 | 63.2 | 8.6 | 81 | 1.32 | 18.1 | 121.5 |
| | Makkosház | 1.3 | 9416 | 20.6 | 27.6 | 54.5 | 17.9 | 111 | 1.39 | 15.9 | 74.6 |
| | Felsőváros II. | 0.6 | 9230 | 11.1 | 32.6 | 56.2 | 11.2 | 92 | 1.37 | 18.6 | 100.0 |
| | Tarjánváros | 0.7 | 13,449 | 11.1 | 28.9 | 64.9 | 6.1 | 75 | 1.31 | 16.0 | 115.7 |
| Kecskemét | Széchenyiváros 1 | 0.7 | 8500 | 14.4 | 34.2 | 58.9 | 7.0 | 72 | 1.27 | 20.6 | 148.6 |
| | Széchenyiváros 2 | 0.2 | 3200 | 4.7 | 7.4 | 88.9 | 3.7 | 103 | 1.67 | 23.3 | 125.0 |

**Table 2.** Main characteristics of the nonHungarian housing estates studied.

| City | Housing Estate | Area [sqkm] | Population | Network Length [km] | Ratio of Dead-Ends [%] | Ratio of Three-Way Intersections [%] | Ratio of Four-Way Intersections [%] | Average Road Length [m] | Connectivity Index | Road Network Density [km/sqkm] | Intersection Density [Intersection/sqkm] |
|---|---|---|---|---|---|---|---|---|---|---|---|
| Linz | Ennsfeld | 0.2 | 3120 | 4.0 | 29.6 | 48.1 | 22.2 | 115 | 1.37 | 17.5 | 82.6 |
| | Heinrich-Kandl Weg | 0.2 | 2507 | 3.2 | 25.0 | 75.0 | 0.0 | 133 | 1.30 | 18.8 | 88.2 |
| Salzburg | Mülln | 0.2 | 1023 | 2.1 | 40.0 | 60.0 | 0.0 | 81 | 1.30 | 11.6 | 66.7 |
| Innsbruck | Neurum | 0.5 | 5522 | 4.3 | 29.2 | 62.5 | 8.3 | 131 | 1.38 | 9.6 | 37.8 |
| | Reichenau | 0.8 | 12,133 | 7.8 | 27.4 | 67.7 | 4.8 | 97 | 1.29 | 9.7 | 56.3 |
| Liberec | Rochlice | 0.3 | 5710 | 5.6 | 37.7 | 49.1 | 13.2 | 73 | 1.25 | 20.0 | 121.4 |
| | Aloisina vysina | 0.2 | 1309 | 4.4 | 44.4 | 52.8 | 2.8 | 100 | 1.14 | 20.1 | 90.9 |
| Olomouc | Foerstrova | 0.4 | 5356 | 4.3 | 48.6 | 45.9 | 5.4 | 98 | 1.19 | 10.2 | 45.2 |
| | F1 (nova ulice) | 0.2 | 3736 | 2.6 | 57.1 | 38.1 | 4.8 | 99 | 1.14 | 11.2 | 39.1 |
| Pilsen | Bory, Jižní Předměstí | 0.4 | 5894 | 6.7 | 34.7 | 62.6 | 2.7 | 78 | 1.17 | 18.0 | 113.5 |
| | Doubravka | 0.2 | 2873 | 2.8 | 34.5 | 65.5 | 0.0 | 95 | 1.21 | 14.2 | 70.0 |
| Kosice | KVP | 0.8 | 13,345 | 11.8 | 29.9 | 68.7 | 1.5 | 117 | 1.40 | 15.8 | 62.7 |

Figures 5–11 show the OSM details of the selected areas, with information on basic land-use. Although the relationship between land-use and network structure is beyond the scope of this paper, these figures can serve as a basis for further analysis.

At first glance, the site with the lowest ratio of dead-ends and the highest ratio of three-way intersections looks like a gridiron structure (Figure 5), but upon closer inspection, almost all four-way intersections change to two staggered three-way intersections.

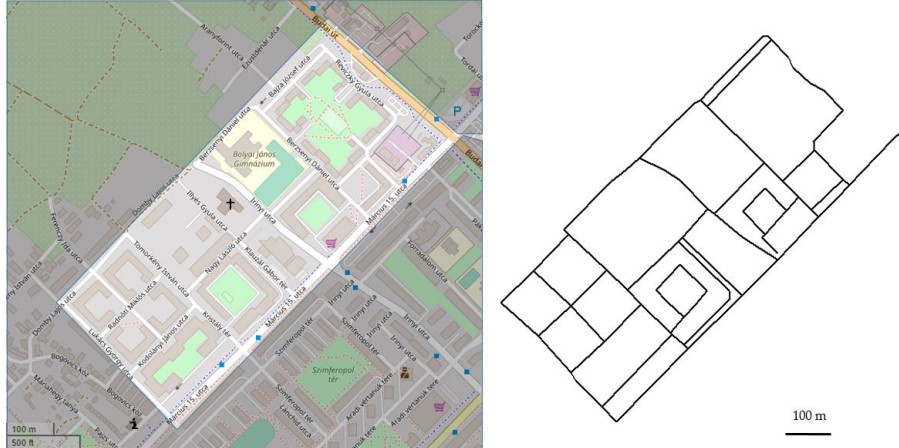

**Figure 5.** Map Map of the housing estate Széchenyiváros 2. (Kecskemét, Hungary).

The site having the highest ratio of dead-ends (50%) and the lowest ratio of three-way intersections is located in Olomouc (Czech Republic) (Figure 6). The former, Kecskemét, is a good example of a highly connected network (connectivity index of 1.67), while the latter was designed according to the Radburn principle in order to separate motorised and nonmotorised traffic, thereby resulting in a high share of dead-ends.

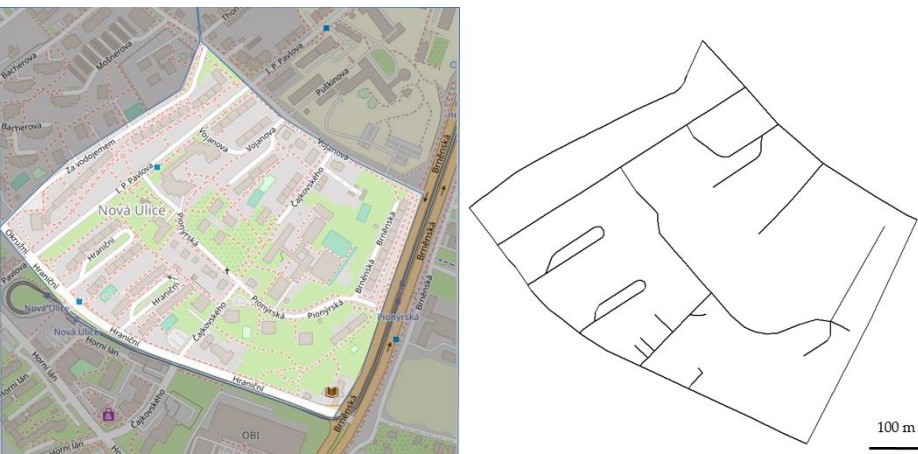

**Figure 6.** Map of the housing estate F1 - nova ulice (Olomouc, Czech Republic).

The highest share of four-way intersections was observed in Linz (Figure 7), with a medium share of three-way intersections and dead-ends. The long, straight, and continuous streets could lead to high speeds unless traffic calming measures are introduced.

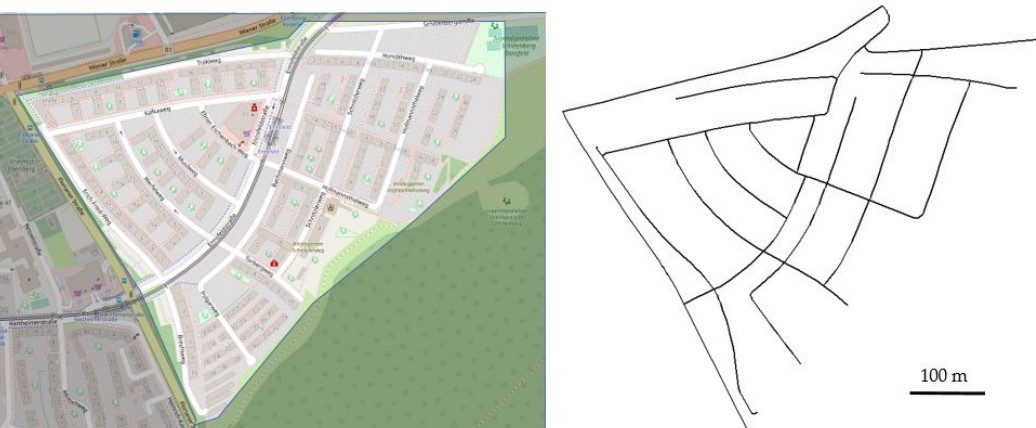

**Figure 7.** Map of the housing estate Ennsfeld (Linz, Austria).

The housing estate in Debrecen, Hungary, which is bordered on three sides by main roads, has no four-way intersections. The internal road network is mainly used by local traffic (Figure 8).

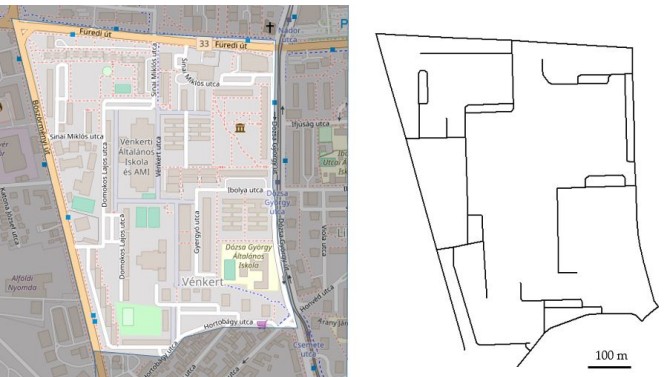

**Figure 8.** Map of the housing estate Vénkert (Debrecen, Hungary).

The road networks of these housing estates show different patterns over different decades. Using the indicators mentioned above, such as the ratio of dead-ends and the

connectivity index, it is possible to quantify the differences. Some examples given here may further illustrate these patterns.

Road networks of the 1950s were typically characterised by network traversability. In housing estates built during this period, we can observe that a major road was designed to cross the estate. A good example is shown in Figure 9, where this link now serves as a four-lane arterial. At that time, the low level of motorisation did not justify the need for such a road to bypass the estate, so it was integrated into the network.

Parking facilities are also part of the road network. In the 1950s, however, parking was not an issue due to the very low level of motorisation. Most vehicles parked parallel to the road, and the networks did not have dedicated parking areas.

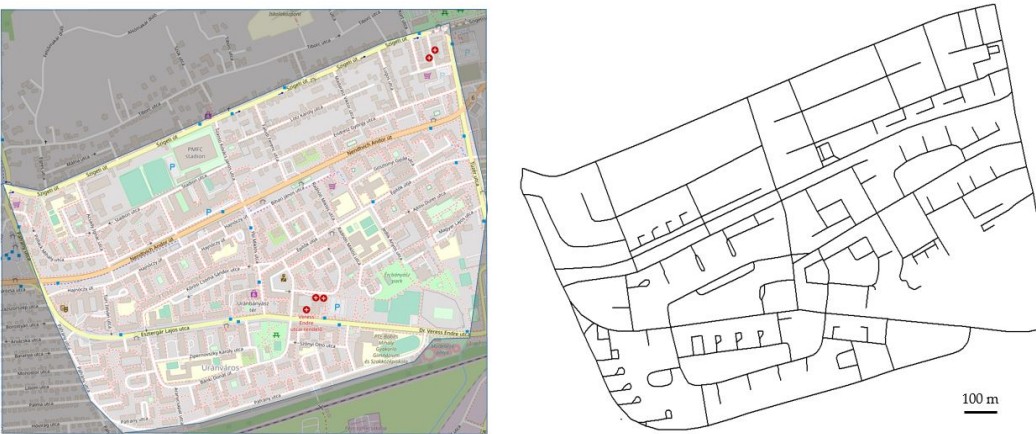

**Figure 9.** Map of the housing estate Uránváros (Pécs, Hungary).

In the 1960s and 1970s, the ratio of dead-ends in housing estates increased significantly. As the traversability of the road network decreased; they started to provide parking lots next to the buildings. The ratio of dead-ends reached its maximum value in the 1970s (see an example in Figure 10). In many Hungarian housing estates, garages were located on the ground floor of the buildings, which was unique in Hungarian cities.

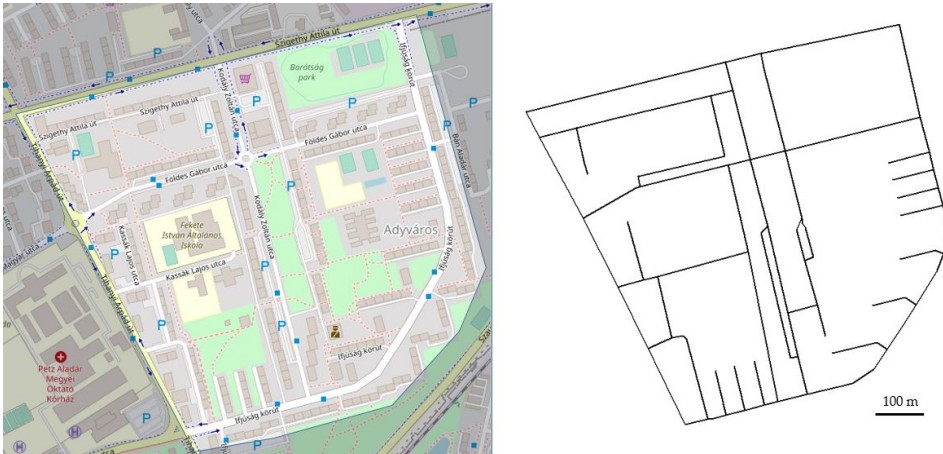

**Figure 10.** Map of the housing estate Adyváros (Győr, Hungary).

Interestingly, road network design in the 1980s echoes that of the 1950s. The main difference, however, is that major roads bypass housing estates. A good example is Marcalváros II. (Győr, Hungary), where the main roads are close to the housing estate, and the internal road network serves local traffic needs (Figure 11). Another difference between the two decades is that, in the 1980s, large parking lots were designed and built to accommodate the rapidly growing number of vehicles. Parking areas had multiple connections to the road network, and this resulted in a lower number of dead-ends. Connections were mostly three-way inter-

sections. These parking lots occupy large spaces and compete with green and recreational areas; there are now initiatives to reduce the area they cover by constructing multistorey parking garages.

There is no direct information available on the costs of these road networks. Economic considerations obviously influenced the network length. However, it is clear that the accessibility of these buildings by car has always been guaranteed. The case is different for parking areas. As the level of motorisation was low at the time of their construction, economic considerations led to the provision of fewer parking facilities, which were later extended by converting green areas into parking lots.

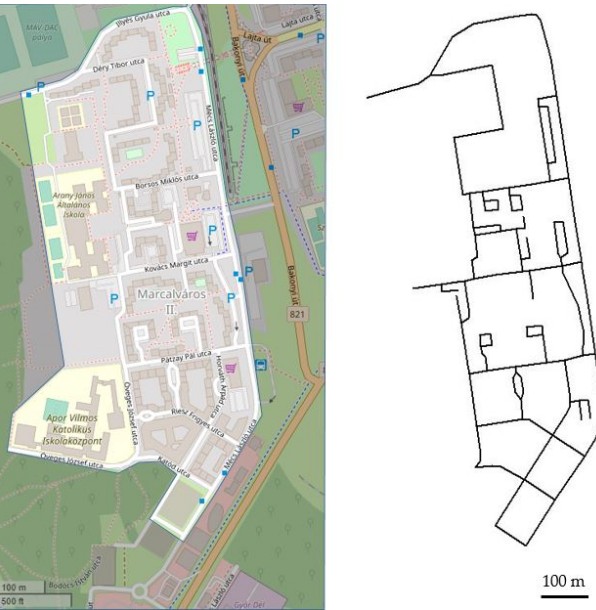

**Figure 11.** Map of of the housing estate Marcalváros II. (Győr, Hungary).

The above reasoning leads us to conclude that the design principles of housing estates have changed over time. These patterns can be seen in the changes in the ratio of dead-ends and traversability of the road network. This is examined in the next section.

## 4. Results

This section analyses the road networks of the selected Central European housing estates. Firstly, the ratio of intersections and dead-ends is investigated, and, secondly, housing estates are compared on the basis of their connectivity index.

### 4.1. Characteristics of Housing Estates Based on Their Nodes

The share of different intersection types in the four basic network patterns is shown in Table 3. The historical network (type A) has a relatively balanced structure among the three intersection types. The gridiron (type B) has no dead-ends, and all intersections are either three-way or four-way intersections. The general network (type C) consists mainly of three-way intersections, with very few dead-ends and four-way intersections, while the suburban network (type D) excludes four-way intersections and has an equal share of three-way intersections and dead-ends.

Triplots [29] can be used to illustrate the ratio of three-way intersections, four-way intersections, and dead-ends. Figure 12 shows the data for all the housing estates selected: blue—Hungarian; grey—Austrian; red—Czech; and yellow—Slovak housing estates; A, B, C and D are the four basic road patterns according to [15] illustrated in Figure 1.

With the exception of one of the basic road patterns, the gridiron indicated with B, all points are clustered around the same region. The share of three-way intersections ranges

from 40 to 90%, and the share of dead-ends ranges from 10 to 60%. Four-way intersections have the lowest share between 10 and 30%.

**Table 3.** Share of different intersection types in the basic network patterns' (rounded values).

| Network Type | Ratio of Three-Way Intersections [%] | Ratio of Four-Way Intersections [%] | Ratio of Dead-Ends [%] |
|---|---|---|---|
| A | 60 | 20 | 20 |
| B | 50 | 50 | 0 |
| C | 85 | 5 | 10 |
| D | 50 | 0 | 50 |

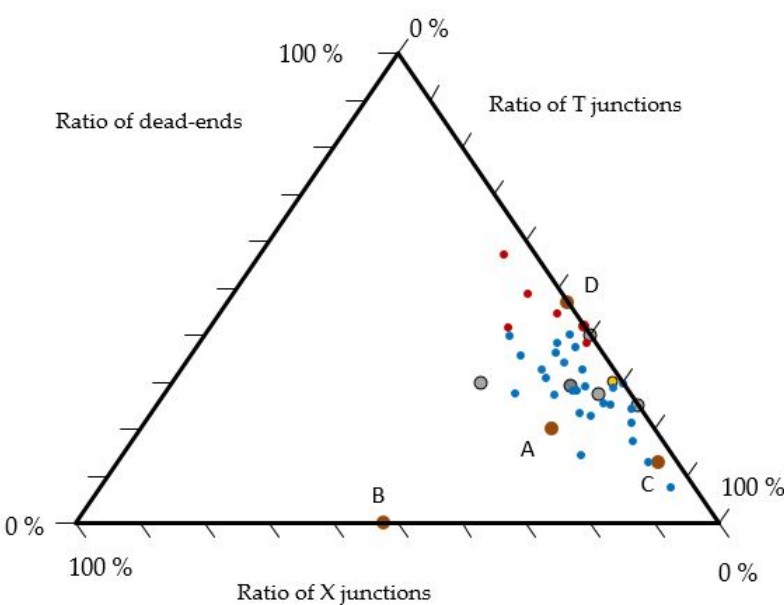

**Figure 12.** Ratio of three-way intersections, four-way intersections and dead-ends of housing estates in selected Central European countries.

The pairwise comparison of the three node types revealed a close correlation and inverse relationship between the ratio of dead-ends and three-way intersections (Figure 13).

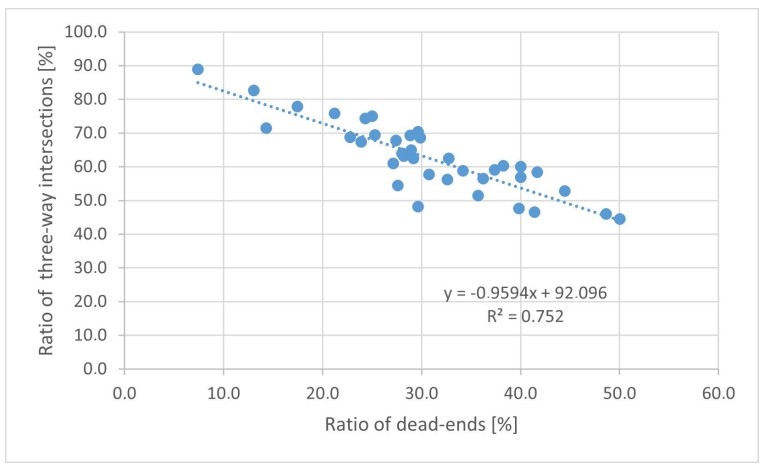

**Figure 13.** Ratio of dead-ends vs. ratio of three-way intersections.

There was no correlation found for the ratio of dead-ends to four-way intersections (Figure 14). This is because dead-ends are usually connected to a three-way intersection and much less often to a four-way intersection.

No correlation was found between the ratio of dead-ends and four-way intersections. In general, four-way intersections are rare within these housing estates; there are some districts where the number of four-way intersections is zero.

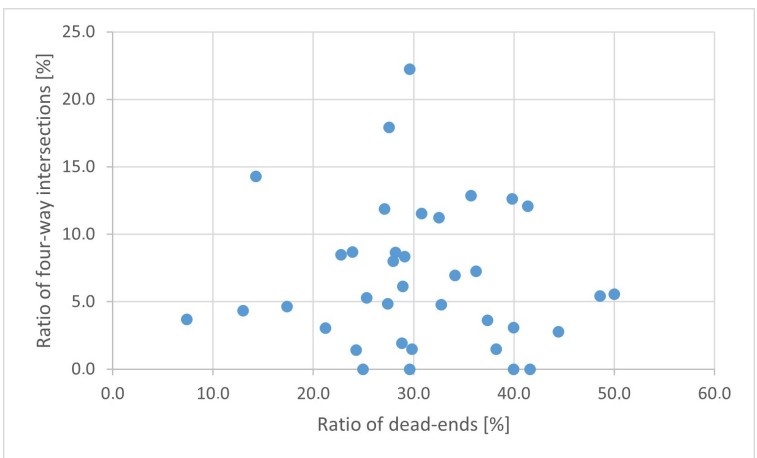

**Figure 14.** Ratio of dead-ends vs. ratio of four-way intersections.

Table 4 shows the composition of three-way intersections, four-way intersections, and dead-ends at their minimum and maximum values for all estates. The lowest ratio of dead-ends had the highest ratio of three-way intersections, and vice versa. These extreme cases are road networks that are either typically dominated by dead-ends or are highly traversable. Of course, blended designs are also present in these housing estates.

**Table 4.** Housing estates with the minimum and maximum ratio of dead-ends, three-way intersections, and four-way intersections.

| City | Ratio of Dead-Ends [%] | Ratio of Three-Way Intersections [%] | Ratio of Four-Way Intersections [%] |
|---|---|---|---|
| Kecskemét—Széchenyiváros 2 | 7.4 (min) | 88.9 | 3.7 |
| Olomouc—F1 | 57.1 | 38.1 (min) | 4.8 |
| Debrecen—Vénkert | 29.6 | 70.4 | 0.0 (min) |
| Olomouc—F1 | 57.1 (max) | 38.1 | 4.8 |
| Kecskemét—Széchenyiváros 2 | 7.4 | 88.9 (max) | 3.7 |
| Linz—Ennsfeld | 29.6 | 48.2 | 22.2 (max) |

Four cases with either the minimum or the maximum value represent two housing estates. The one having the lowest ratio of dead-ends (7.4%) and the highest ratio of three-way intersections (88.9%) are both located in Kecskemét, Hungary. Conversely, the F1 housing estate in Olomouc has the highest ratio of dead-ends (57.1%) and the lowest ratio of three-way intersections (38.1%).

### 4.2. Analysis of the Connectivity Index

The connectivity index is one of the most widely used indicators to measure the traversability of road networks (Equation (1)). The boxplots show the evolution of the connectivity index separately for Hungary (Figure 15), as well as for Slovakia, the Czech Republic, and Austria together (Figure 16). In this plot, horizontal lines show the minimum, first quartile, median, third quartile, and maximum, while small circles are considered as outliers. The reason for separating the countries was to see if Hungary showed any significant differences compared to the neighbouring countries and also because of the nature of the samples as described under Section 3.

The two boxplots show that the trend in traversability over time has been very similar. The less traversable road network patterns of the 1960s and 1970s transformed into more traversable ones in the 1980s. This is much more visible in Hungarian estates.

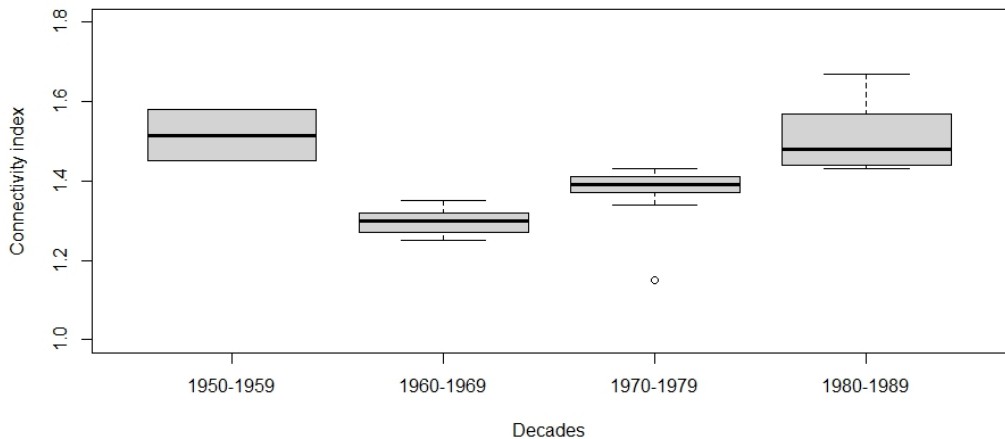

**Figure 15.** Connectivity of the housing estates in Hungary over decades.

In the case of nonHungarian housing estates, there are only three boxplots. There was only one housing estate built in the 1950s (Innsbruck, Reichenau), which was merged with those built in the 1960s, as well as because its connectivity index was not significantly different. Hungarian housing estates built in the 1950s and 1960s showed significant differences (1.45–1.58 and 1.25–1.35, respectively) and were therefore not merged.

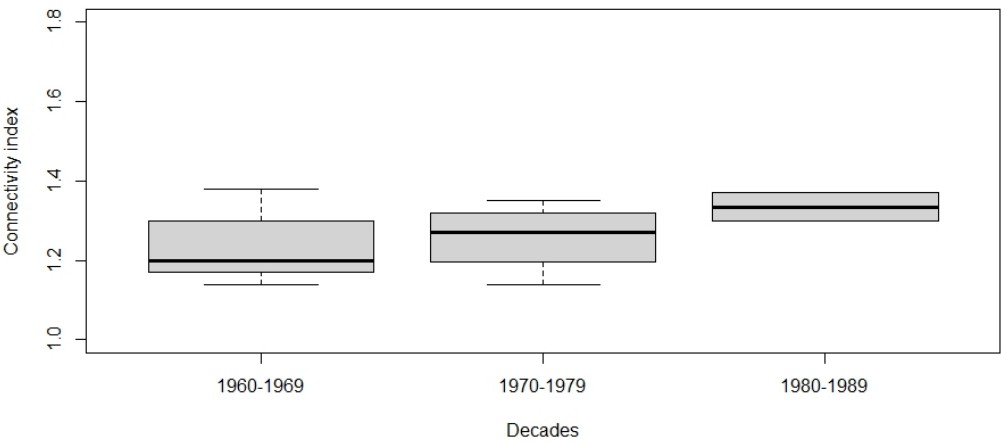

**Figure 16.** Connectivity of the housing estates in Slovakia, the Czech Republic, and Austria over decades.

A Kruskal–Wallis rank sum test was used to test whether the differences between the decades were significant. The Kruskal-Wallis test is a nonparametric alternative to the one-way ANOVA to test for significant differences between groups. In the case of the Hungarian housing estates, this change over time was found to be significant (Table 5), but it was not significant for the other respective Central European countries.

**Table 5.** Kruskal–Wallis test results.

| Test Results | Hungary | Slovakia, the Czech Republic, and Austria Combined |
|:---:|:---:|:---:|
| $\chi^2$ | 16.1730 | 2.0028 |
| df | 3 | 2 |
| *p*-value | 0.001045 * | 0.3674 |

* significant result.

Pairwise comparisons can be made using the Wilcoxon rank sum test. The results (Table 6) show that there are significant differences for the decades after the 1960s. However, the road networks built in the 1950s are not significantly different from any of the later decades.

**Table 6.** Pairwise comparison of decades for the Hungarian housing estates.

| *p*-Values | 1950–1959 | 1960–1969 | 1970–1979 |
|---|---|---|---|
| 1960–1969 | 0.086 | - | - |
| 1970–1979 | 0.066 | 0.043 * | - |
| 1980–1989 | 0.857 | 0.013 * | 0.013 * |

* significant results.

An interesting question is what are the possible solutions to further improve connectivity and reduce congestion in these estates, and how feasible are these solutions? The authors believe that improving connectivity for cars in these housing estates is not an option, as it would create additional traffic. On the other hand, improving connectivity for cyclists and pedestrians would promote sustainable development. Traffic congestion is most evident on the surrounding main roads. Possible solutions to these problems are beyond the scope of this paper.

## 5. Discussion and Limitations

This paper analyses the road topology of the housing estates in a few selected Central European countries (Hungary, Austria, Czech Republic, and Slovakia). These housing estates are special types of urban quarters, which were built from the 1960s to the 1980s, with an area of 0.2 to 1.5 sqkm, a road network density of 10–20 km/sqkm, and with 4–10-storey buildings, which were mostly planned and implemented by local governments as comprehensive projects, including the site preparation, public utilities, road networks, housing, and other buildings (kindergartens, schools, shops, etc.). The sites were analysed in three steps: (1) the road network topology of different decades from the 1950s to the 1980s was described, (2) the ratio of intersections and dead-ends was examined, and (3) the connectivity indices were analysed and compared. It should be noted that the sample of 37 housing estates is not sufficient to draw conclusions for specific countries or regions. Nevertheless, the general conclusions may be useful.

In Central Europe, housing estates built before the fall of the iron curtain are of a different nature to those built after 1990. This was the main reason for not including newer estates in our analysis. Nevertheless, the authors believe that this could be an interesting direction for future research.

This study focused on network issues and therefore did not consider wider land-use aspects. From experience, it can be stated that housing estates before 1990 in the Eastern Bloc countries had many ten-storey buildings, but space was left within the estates for shops, kindergartens, schools, and green areas. On the other hand, housing estates built after 1990 usually have lower buildings but no shops, schools, and kindergartens. Therefore, residents have to use their cars more often. In a next step, we plan to analyze these issues.

This research has shown that, although there are street network patterns in the literature (such as in [10,15]), the street networks in cities are diverse, and multiple permutations of street patterns exist under the main categories [7].

Safety issues were not addressed in this paper. However, there is evidence in the literature that three-way intersections tend to be safer than four-way ones, and roundabouts are even safer [30–32]. The continuity and self-explaining nature of intersections within housing estates play an important role in safety.

## 6. Conclusions

This research has demonstrated that the design policy for road networks in housing estates has changed over time, with connectivity becoming more important. This change

was driven by the rapid increase in motorisation rates and the need for high levels of connectivity [20,21]. This need has been met to varying degrees in different housing estates, with a greater increase in the connectivity index in Hungarian estates. Further research could reveal the underlying reasons for these policies, namely, how local conditions and urban planning trends have influenced road network topology.

We found that, despite many differences between the sites (e.g., terrain, location within the city, brownfield/greenfield, and local history), some indicators such as size, road section length, and network density are quite similar in most of the housing estates. These indicators were probably not calculated by the planners at the time, but the result is still quite similar, in terms of required building density, free spaces, and accessibility. The similarity is probably due to the similar history and size of these cities and the similar aims of the planners at the time. However, changes over time have also been noted.

Compared to the basic types of network patterns shown in Figure 1, we can conclude that all of the networks analysed are very far from the gridiron structure (B). On the other hand, we can observe a great variety of structures. Some of them are similar to schemes A, C, or D, but most of them fall between these structures (Figure 1).

The trends in the design of road networks over the decades have been similar in the housing estates in the four Central European countries. In all of the cases, they are dominated by three-way intersections, the share of which correlates well with the share of dead-ends. The highest share of dead-ends was found in Hungarian housing estates. Road topologies showed remarkable differences between decades. Statistical tests showed that these differences were significant in Hungary after the 1960s, but they were not significant in the other three countries.

Further research could also look at the network structures of walking and cycling facilities, where connectivity and short distances are even more important. Another potential area of research is the transformation of traditional gridiron networks through local traffic calming measures (e.g., closures and one-way streets) and their impacts on safety.

**Author Contributions:** Conceptualization, C.K. and P.H.; methodology, P.H. and A.B.; software, P.H.; formal analysis, P.H. and C.K.; investigation, P.H. and A.B.; resources, P.H.; writing—original draft preparation, P.H. and A.B.; writing—review and editing, C.K.; visualization, A.B.; supervision, A.B. and C.K. All authors have read and agreed to the published version of the manuscript.

**Funding:** This research received no external funding.

**Data Availability Statement:** Not applicable.

**Conflicts of Interest:** The authors declare no conflict of interest.

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
