# Peer review of "Evolution of the Road Network Topology of Central European Housing Estates"

_infrastructures, doi:10.3390/infrastructures8100142_

Round 1

Reviewer 1 Report (Previous Reviewer 1)

The new version of this topic looks better than the former one, especially from the graphical side. The Authors answered the remarks but only in a letter to the editor (and reviewer). I have not found these answers in the new version of the manuscript. Some answers are not full and satisfactory. Some topics are controversial and discussable. Reviewers and readers wish to know more details and findings about these topics. Such a discussion and commentaries should be composed into the manuscript.

Especially, some broader explanation for the problems collected below is needed in the text.

1. The fall of the iron curtain primarily concerns the “east block” countries, here Hungary, Slovakia, and the Czech Republic; Austria belongs the whole time to the “West”. Were the same processes and differences in the sense of house estate building in Austria according to the rest of the considered countries?

2. The phenomenon of the lack of differences in the road network after the year 1990 should be proven. For example, showing the map (or photos) of the chosen part of the selected city (or cities) with the road network in 1990 and in the present time.

3. The map with the location of all considered cities is necessary. Please not forgot about the names of these cities on the map. The scale will be nice.

4. The number of residents in the estates is not presented. Please, add these data to Tables 1 and 2.

There is no one simple answer for the problems pointed out above. My remark is only on more Author’s commentaries in the text. This will allow an increase in the clarity and readability of the manuscript.

Please attach the file with tracking of the changes when sending the revised version of the manuscript. This will be helpful for the review.

Minor editing of English language required.

Author Response

The new version of this topic looks better than the former one, especially from the graphical side. The Authors answered the remarks but only in a letter to the editor (and reviewer). I have not found these answers in the new version of the manuscript. Some answers are not full and satisfactory. Some topics are controversial and discussable. Reviewers and readers wish to know more details and findings about these topics. Such a discussion and commentaries should be composed into the manuscript.

Thank you for this comment. We added these answers at several points in the manuscript. In order for the reviewers to find these, along with the modified manuscript a manuscript file with changes is also uploaded.

Especially, some broader explanation for the problems collected below is needed in the text.

  1. The fall of the iron curtain primarily concerns the “east block” countries, here Hungary, Slovakia, and the Czech Republic; Austria belongs the whole time to the “West”. Were the same processes and differences in the sense of house estate building in Austria according to the rest of the considered countries?

Thank you for this remark. We added some related text.

“Our sample included housing estates from the former “East Bloc” countries, such as Hungary, Slovakia, and the Czech Republic as well as from Austria which belonged to the “West”. There were undoubtedly differences between these countries in the processes of establishing housing estates; but these differences are beyond the scope of this research. From this sample it is remarkable that Hungary has several large housing estates (over 1 sqkm or 10,000 population), whereas the other three countries with only two exceptions (Innsbruck Reichenau and Kosice KVP) have estates of smaller size. This latter subset of countries was quite limited in terms of sample size.

  1. The phenomenon of the lack of differences in the road network after the year 1990 should be proven. For example, showing the map (or photos) of the chosen part of the selected city (or cities) with the road network in 1990 and in the present time.

Text was added to explain:

“The road networks of the selected housing estates were taken from the OSM database, based on the actual network data at the time of this research. Based on local experiences, it was assumed that the internal network structure of the housing estates did not change over time. The changes related to the roads in these areas were mostly the expansion of parking capacity and traffic engineering measures, such as signalization on the surrounding arterial roads. These changes did not affect the network structure. However, it must be acknowledged that some minor changes may have occurred.”    

  1. The map with the location of all considered cities is necessary. Please not forgot about the names of these cities on the map. The scale will be nice.

We added a figure showing the location of the cities. The scale is indicated in the bottom right corner of the figure.

  1. The number of residents in the estates is not presented. Please, add these data to Tables 1 and 2.

We added population data to both tables.

There is no one simple answer for the problems pointed out above. My remark is only on more Author’s commentaries in the text. This will allow an increase in the clarity and readability of the manuscript.

All these comments are valuable, we agree that no one simple answer can address them. We added our reasonings to the manuscript.

Reviewer 2 Report (New Reviewer)

This study analyzed whether there are big differences in the road topologies in four central European countries between decades. 

As the author stated in the conclusion, I agree with the opinion that the road topology in housing estates change with the passage of time.  

Therefore, in terms of methodology, it is not clear whether it was necessary to distinguish between Hungarian housing estates and non-Hungarian housing estates.  

I wonder what the results will be if housing estates or road networks were classified and compared by the era in which they were created. 

Furthermore, I think there ate not enough samples from a specific countries or regions for comparative analysis by country or region. 

The quality of English is judged to be sufficient. 

Author Response

This study analyzed whether there are big differences in the road topologies in four central European countries between decades. 

As the author stated in the conclusion, I agree with the opinion that the road topology in housing estates change with the passage of time.  

Therefore, in terms of methodology, it is not clear whether it was necessary to distinguish between Hungarian housing estates and non-Hungarian housing estates.

We agree, but due to the limited sample size in the other three countries, and a reasonable sample in Hungary, we kept this separation.   

I wonder what the results will be if housing estates or road networks were classified and compared by the era in which they were created. 

We were only able to show the differences in terms of connectivity index, and this index showed significant changes for Hungary, but not for the other countries. This is due to the differences among countries in sample size 

Furthermore, I think there are not enough samples from a specific countries or regions for comparative analysis by country or region. 

Text was added.

“It has to be mentioned that the sample of 37 housing estates is not enough to draw conclusions for specific countries of regions. Still, the general conclusions might be useful.”

Reviewer 3 Report (New Reviewer)

The topic presented in the paper is on time. In the reviewed paper, the Authors presented the evolution of the road network topology of Central European housing estates. In this study, the Authors presented the road topology of housing estates in a few selected Central European countries (Hungary, Austria, Czech Republic, Slovakia) is analyzed. This research was carried out in three steps: the road network topology of different decades from the 50s to the 80s was described, then, the ratio of intersections and dead-ends was investigated, and at the end, connectivity indices were analyzed and

compared. The research was carried out using ESRI ArcGIS software. In my opinion, the paper can be considered for publication, after taking into account the following remarks:

- in the "Keywords" section, the keyword "transport infrastructure" should be added,

- in the Introduction section, the Authors should mention, how important is to design safe solutions in housing estates. The key roles play design such solutions like roundabouts, because this kind of intersection lead to small vehicles' speed ("The Comparison of Models for Critical Headways Estimation at Roundabouts", doi 10.1007/978-3-319-43985-3_18) and they have many other advantages leading to aesthetic solutions("Roundabouts as aesthetic road solutions for organizing landscapes", doi 10.20858/sjsutst.2022.115.4). The Authors should mention about this roundabouts properties and refer to the mentioned above research which these properties was described. One short paragraph in the Introduction section will be enough,

- below figure 1, the Authors wrote as follows "Figure 1. Road network patterns according to Marshall [15]." Do the Authors have written permission from [15] for further use of the figure? Usually publishing houses ask for such written permission,

- at the end of the Introducion section, the Authors wrote the main aim of the paper. It is very good, but after that, should be shortly written what was contained in each paper section,

- in the figure called "Figure 3. Node marked with a red circle is a dead-end on macro level and a three-way intersection on micro level ." the legend should be added because the reader doesn't know what does it mean the red and blue circles,

- the section called "5. Discussion and conclusions" is underdeveloped and should be extended by adding some reference to the other similar research.

English seems to be improved.

Author Response

- in the "Keywords" section, the keyword "transport infrastructure" should be added,

Thank you, it is indeed a relevant keyword, it has been added.

- in the Introduction section, the Authors should mention, how important is to design safe solutions in housing estates. The key roles play design such solutions like roundabouts, because this kind of intersection lead to small vehicles' speed ("The Comparison of Models for Critical Headways Estimation at Roundabouts", doi 10.1007/978-3-319-43985-3_18) and they have many other advantages leading to aesthetic solutions("Roundabouts as aesthetic road solutions for organizing landscapes", doi 10.20858/sjsutst.2022.115.4). The Authors should mention about this roundabouts properties and refer to the mentioned above research which these properties was described. One short paragraph in the Introduction section will be enough,

The authors agree with the favorable characteristics of roundabouts. A paragraph on the safety of roundabouts have been added to the discussion section.

- below figure 1, the Authors wrote as follows "Figure 1. Road network patterns according to Marshall [15]." Do the Authors have written permission from [15] for further use of the figure? Usually publishing houses ask for such written permission,

The authors have not asked for permission since the original work where the figure was published is cited. However, the authors will double check that in case of publishing a permission is needed or not.

- at the end of the Introducion section, the Authors wrote the main aim of the paper. It is very good, but after that, should be shortly written what was contained in each paper section,

Thank you, this has been added.

- in the figure called "Figure 3. Node marked with a red circle is a dead-end on macro level and a three-way intersection on micro level ." the legend should be added because the reader doesn't know what does it mean the red and blue circles,

Figure 3 now contains legend.

- the section called "5. Discussion and conclusions" is underdeveloped and should be extended by adding some reference to the other similar research.

The authors decided to split this last section into “Discussion and limitations” and “Conclusions”. Further elaboration and several more literature sources have been added.

Round 2

Reviewer 1 Report (Previous Reviewer 1)

The answers, comments, and supplements are satisfactory. I think this manuscript version can be published with no additional changes. This is an interesting study and appropriate for publishing in "Infrastructures".

Minor editing of English language required.

Author Response

Thank you for your positive feedback.

Reviewer 2 Report (New Reviewer)

New version of the manuscript are not full and satisfactory for my question. In particular, no modifications have been made to the methodology. 

Author Response

To a certain extent we agree with the previous comments of this reviewer. We stressed that our sample has certain limitations, and, in our view, it is not a question of methodology. Nevertheless, to better expose these limitations we decided to split the last section of the manuscript into “Discussion and limitations” and “Conclusions”.

Round 3

Reviewer 2 Report (New Reviewer)

Unfortunately, I cannot find any improvements in the revised version. 

This manuscript is a resubmission of an earlier submission. The following is a list of the peer review reports and author responses from that submission.

Round 1

Reviewer 1 Report

The topic is interesting but the range of analyses and presentation are too poor. The consideration of publishing should be decided after the explanations, complementation and corrections below. 

1. Why this analysis concerns only the 1960 – 1990 period? Newer estates do not exist? Are these not worth analysis?

2. Accepting the limitation of the study period the next question is important. I think that today’s road networks were considered. Why? Were these networks not changed between 1990 and 2023 in all considered locations?

3. The more general map (showing Central Europe: Hungary, Austria, Czech Republic, Slovakia) with locations of considered cities is needed.

4. The selection of housing estates is not clear. Are these estates really similar? There is a lack of important data about the estates: population (number and density), number (and ratio) of “multi-storey housing estates” (this term, line ??? on page 2 is not defined, how many floors were considered?), year (or decade) of construction, the ratio of brownfield/greenfield. The manuscript contains no line numbering.

5. Tables 1 and 2 are too fast presented. Some data are not understandable (for example: “connectivity index”). This measure is defined on page 5 but should be defined before the presentation of Tables 1 and 2.

6. The scale on maps of housing estates (figures 7 – 9 and 11 – 13) with lines representing appropriate distance (for example 300 m) is needed. Why this part of the study contains only 7 estates? What was a selection way to choose these estates?

7. The discussion part should be enhanced considering the enlarged set of data characterizing the estates (see remark 4).

Minor editing of English language required.

Reviewer 2 Report

Manuscript ID: infrastructures-2406201

Title: Evolution of the road network topology of Central European housing estates

This paper analyzes the road network topology of housing estates in Central European countries, including Hungary, Austria, Czech Republic, and Slovakia. The study finds that connectivity has become increasingly important in the design of road networks over time. The authors suggest that further research could reveal the underlying reasons behind these policies and how local conditions and trends in urban planning influenced road network topology. The paper also discusses the origins of road network topology analysis and its significance in transportation infrastructure. Overall, the study provides insights into the evolution of road networks in Central European housing estates and highlights potential implications for urban planning and transportation policy. It is suggested that this paper can be recommended for publication after a minor revision. The specific reasons are listed below:

l  Abstract and introduction are too long, it is suggested to significantly shorten them in a well-organized fashion. Moreover, the literatures are not state-of-the-art or state-of-the-practice, recent publications should be added to strengthen the up-to-date literatures.

l  The font size and of texts, figures and tables should be consistent. For example, the text of main body is Palatino Linotype and the text type of figures is of the other type.

l  The innovative contribution of this article should be well elaborated. The contribution seems to be incremental so that the authors should clarify their contributions compared to the past work.

Moreover, the following questions should also be elaborated:

l  How do cultural and historical factors influence the design of road networks in Central European housing estates, and how have these factors changed over time?

l  To what extent do economic considerations, such as the cost of construction and maintenance, impact the design of road networks in these housing estates?

l  How can emerging technologies, such as autonomous vehicles and smart transportation systems, be integrated into existing road networks in Central European housing estates?

l  What are some potential solutions for improving connectivity and reducing traffic congestion in these housing estates, and how feasible are these solutions from a practical standpoint?

l  How do different types of junctions (e.g. roundabouts, signalized intersections) impact traffic flow and safety in Central European housing estates, and what are some best practices for designing effective junctions?

The english writting is acceptable